# Evaluation of QOL in Patients with Dyspeptic Symptoms Who Meet or Do Not Meet Rome IV Criteria

**DOI:** 10.3390/jcm11010021

**Published:** 2021-12-22

**Authors:** Masatoshi Mieno, Toshihiko Tomita, Sota Aono, Katsuyuki Tozawa, Keisuke Nakai, Takuya Okugawa, Masashi Fukushima, Tadayuki Oshima, Hirokazu Fukui, Hiroto Miwa

**Affiliations:** 1Division of Gastroenterology and Hepatology, Department of Internal Medicine, Hyogo College of Medicine, Nishinomiya 663-8501, Japan; ma-mieno@hyo-med.ac.jp (M.M.); tomita@hyo-med.ac.jp (T.T.); so-aono@hyo-med.ac.jp (S.A.); k-nakai@hyo-med.ac.jp (K.N.); okugawat@hyo-med.ac.jp (T.O.); ma-fukushima@hyo-med.ac.jp (M.F.); t-oshima@hyo-med.ac.jp (T.O.); hfukui@hyo-med.ac.jp (H.F.); 2Department of Gastroenterology and Hepatology, Amagasaki Chuo Hospital, Amagasaki 661-0976, Japan; katu-you@chuoukai.or.jp

**Keywords:** Rome IV criteria, functional dyspepsia, subdivision of non-FD, diagnosis

## Abstract

Health related quality of life (HR-QOL) of functional dyspepsia (FD) patients is impaired. However, the QOL of such patients has not been fully examined. Accordingly, we examined the QOL of Rome IV defined FD, endoscopic negative dyspeptic patients who do not meet the criteria, (non-FD patients) and healthy subjects, and investigated the factors that influence HR-QOL. This was a multicenter, prospective, observational study. Two hundred thirty-five patients (126 FD, 87 non-FD) and 111 healthy subjects were investigated, and non-FD patients were subdivided into three groups: 17 patients failing to meet only the disease duration criterion (Group A), 53 patients failing to meet only disease frequency criterion (Group B) and 17 patients failing to meet both the disease duration and frequency criteria (Group C). They completed a questionnaire survey regarding gastrointestinal symptoms (GSRS), QOL and psychological factors, which were compared among three groups. The total GSRS score was significantly higher in FD patients than non-FD patients (*p* = 0.012), which was higher than the healthy subjects (*p* < 0.0001). Furthermore, the total GSRS score of FD patients was comparable to that of Group A (*p* = 0.885), which was significantly higher than that of the Group B and C (*p* = 0.028, *p* = 0.014, respectively). HR-QOL is more impaired in FD patients than non-FD patients, which was significantly lower than the healthy subjects. That GSRS score in FD and Group A was comparable suggesting that an increased frequency of symptoms may have impact on the impairment of patient’s QOL.

## 1. Introduction

Functional dyspepsia (FD) is a disorder that presents with dyspepsia without organic diseases [1,2,3,4]. The Rome criteria are frequently and widely used to define FD, but patients with dyspeptic symptom do not always meet the Rome IV criteria. The Rome IV criteria defines FD as having one or more of the four symptoms (postprandial fullness, early satiation, epigastric pain, and epigastric burning) for over six months, with symptoms persisting for the previous three months. A condition that involves the former two of these four symptoms is called postprandial distress syndrome (PDS), which requires symptom frequency for at least three days per week. A condition that involves the latter two is called epigastric pain syndrome (EPS), which requires symptom frequency for at least one day per week [5,6,7,8,9].

It has been reported that only about half of the patients visiting hospitals meet the Rome criteria [10] in a real-world clinical setting. Recent reports in Asia also indicate that the Rome criteria for FD is suitable only in certain cases [11,12,13,14,15,16], meaning there are many dyspeptic patients who do not meet the Rome criteria, which we call non-FD patients in this study.

Nevertheless, health related quality of life (HR-QOL) of non-FD patients has not been well documented. Therefore, in the present study, we examined and compared HR-QOL of such patients as well as the gastrointestinal symptoms, psychological factors (such as anxiety and depression) to investigate the clinical significance of non-FD patients and usefulness of the Rome IV criteria.

## 2. Materials and Methods

### 2.1. Study Design

This was a multicenter, prospective, observational study conducted jointly at our and affiliated hospitals (eight centers). Between April 2020 and April 2021, a questionnaire survey on gastrointestinal symptoms, psychological factors, and HR-QOL was conducted in 235 patients who had visited the outpatient clinic complaining of dyspeptic symptoms of four symptoms of Rome IV criteria, and 111 healthy subjects without symptoms (control group). The healthy subjects were recruited among the subjects who visited the hospital for the medical check. The organic diseases had been excluded from these dyspeptic patients by esophagogastroduodenoscopy (EGD). All the patients and healthy subjects had provided written informed consent to participate in this study. Approval was obtained from the ethics committee at Hyogo College of Medicine (Approval No. 3467; University Hospital Medical Information Network registration number UMIN000041094). The trial was conducted according to the principles governing human research in the Declaration of Helsinki. All authors had access to the study data and reviewed and approved the final manuscript.

### 2.2. Patients

The inclusion criteria for patients included the following: (1) outpatients aged 20–75 years; (2) patients with upper abdominal symptoms (e.g., postprandial fullness, early satiation, epigastric pain, and epigastric burning); (3) patients with EGD performed within one year after medical consultation and not showing evidence of organic disease (e.g., malignancy, peptic ulcer, or esophagitis) that causes upper abdominal symptoms; (4) patients not taking antidepressants, anxiolytics, or antipsychotics; and (5) patients who understood the content of this study and provided written, informed consent to participate in the research.

The exclusion criteria included the following: (1) patients with apparent causes of upper abdominal symptoms such as malignancy, peptic ulcer, and systemic disease (e.g., neurological disorders such as Parkinson’s disease or metabolic diseases such as diabetes mellitus); (2) patients with predominance of gastroesophageal reflux disease (GERD) diagnosed based on the Los Angeles (LA) classification; (3) patients with predominance of irritable bowel syndrome (IBS) diagnosed based on the Rome IV criteria; (4) patients who had previously undergone upper gastrointestinal surgery of the stomach, esophagus, etc. (all endoscopic surgery such as endoscopic mucosal resection and endoscopic submucosal dissection were acceptable); (5) patients with severe hepatic or renal impairment; (6) patients with concomitant or suspected psychiatric disorders; and (7) patients who were considered inappropriate for inclusion in the study for other reasons such as poor performance status or poor general condition.

### 2.3. Assessment

FD was diagnosed using the Rome IV criteria [5]. The clinical characteristics were examined and compared between the three groups (dyspeptic patients who met Rome IV criteria, those who did not meet Rome IV criteria and healthy subjects). The items examined included age, sex, body mass index (BMI), smoking history, drinking history, history of *Helicobacter pylori* (*H. pylori*) eradication, and type of their symptoms. *H. pylori* infection was confirmed by serum *H. pylori* antibody or urea breath test.

Non-FD group was subdivided into three groups: (1) patients failing to meet only the disease duration criterion (Group A); (2) patients failing to meet only the disease frequency criterion (Group B); and (3) patients failing to meet both the disease duration and frequency criteria (Group C). 

For all the groups, the clinical characteristics, the Gastrointestinal Symptom Rating Scale (GSRS), the Hospital Anxiety and Depression Scale (HADS), and the 8-item Short-Form Health Survey (SF-8) were examined and compared. 

### 2.4. Questionnaires (Digestive Symptoms, Psychological, HR-QOL)

Gastrointestinal symptoms were assessed using the Japanese version of the GSRS questionnaire, which is described elsewhere [17]. Briefly, the GSRS consists of 15 items. Each belongs to one of five subscales (reflux, abdominal pain, dyspepsia, diarrhea, and constipation). The options for each item range from one to seven. The mean value for each item was used as the score for the subscale, and the mean value for all the subscales was used as the overall score [18,19].

The HADS was used as the anxiety and depression scale, which is described elsewhere [20,21]. Briefly, the HADS consists of 14 items, seven each for anxiety and depression. It assesses these two conditions and provides a total score. Higher scores indicate higher psychological distress [22].

The SF-8 was used to assess HR-QOL. The SF-8 measures eight health concepts: physical functioning (PF), daily role physical (RP), body pain (BP), general health (GH), vitality (VT), social functioning (SF), daily role emotional (RE), and mental health (MH) [23,24].

### 2.5. Stastical Analysis

All results are expressed as means ± standard deviation (SD). One-way ANOVA, the paired t-test, Mann-Whitney U test, and Fisher’s exact test were used and the significance was defined as *p* < 0.05. The statistical analysis was performed using GraphPad Prism 5 (GraphPad Software, La Jolla, CA, USA).

## 3. Results

### 3.1. Enrolment of the Patients

Two hundred thirty-five patients whose chief complaint was dyspepsia (166 patients from university hospital, and 69 patients from eight affiliated hospitals), and 111 healthy subjects who visited the hospital for medical check were administered self-reported questionnaires (GSRS, HADS, SF-8).

Of 235 dyspeptic patients, five were excluded for the positive test for *H. pylori.* and 17 patients who reported no symptoms on GSRS questionnaire survey were also excluded, which have the final analysis included 213 patients. Among these, 126 (59%) met the Rome IV criteria, and 87 (41%) did not. The patient flow is summarized in Figure 1.

### 3.2. Baseline Characteristics and Symptoms

The background characteristics of FD group (EPS, PDS, and overlap), non-FD group and control group are shown in Table 1. The five groups were not significantly different in terms of the following background factors: age, gender, BMI, history of smoking, history of drinking alcohol and proportion of post-*H. pylori* patients. 

### 3.3. Subdivision of Non-FD Patients

The 87 patients in non-FD group were subdivided into three groups. There were 17 (20%), 53 (60%), and 17 (20%) patients in Groups A, B, and C, respectively (Figure 2), indicating the number of non-FD patients who did not meet the duration of the criteria was 34 (40%) and that of those who did not meet the frequency was 70 (80%).

The background characteristics of the non-FD patients in the three sub-groups, FD patients, and healthy subjects are shown in Table 2. There was not significantly different among five groups.

### 3.4. GSRS Score of the Three Groups (FD, Non-FD and Control)

The mean GSRS score of FD (2.8 ± 1.0) was significantly higher than that of non-FD (2.5 ± 0.9), which was significantly higher than that in the control group (1.8 ± 0.6). (Figure 3a). Each domain of the GSRS score (reflux, abdominal pain, dyspepsia, diarrhea, and constipation) is shown in the Figure A1. Here, the mean GSRS scores for the abdominal pain and dyspepsia were 3.2 ± 1.3 and 2.8 ± 1.2, respectively, which were significantly higher than those of non-FD group (2.6 ± 1.3 and 2.3 ± 1.0, respectively, *p* < 0.001), which were also significantly higher than those of the control group (1.6 ± 0.8 and 1.7 ± 0.7, respectively, *p* < 0.0001).

### 3.5. HADS Score of the Three Groups (FD, Non-FD and Control)

The total HADS score of FD (16.4 ± 7.0) was significantly higher than that of non-FD (13.7 ± 6.6), which was significantly higher than that in the control group (11.4 ± 5.9) (Figure 3b). The HADS score of the two domains (anxiety and depression) was shown in the Figure A2.

### 3.6. SF-8 Score of the Three Groups (FD, Non-FD and Control)

The SF-8 physical component summary (PCS) score of FD (43.1 ± 8.5) was significantly lower than that of non-FD (45.6 ± 6.9), which was significantly lower than that in the control group (50.5 ± 5.0) (Figure 3c). The mental component summary (MCS) score was comparable between FD (41.9 ± 8.7) and non-FD group (44.0 ± 7.6), which was significantly lower than that of control (49.4 ± 6.7) (Figure 3d).

### 3.7. Comparison of Clinical Characteristics among the Four Groups (FD Group, Group A, Group B and Group C)

The total mean scores of the GSRS for Group A (2.8 ± 1.2) and FD group (2.8 ± 1.0) were comparable, whereas those of Group B (2.5 ± 0.9) and Group C (2.2 ± 0.6) were significantly lower than of FD group (Figure 4a).

The total mean scores of HADS of FD (16.4 ± 7.0), Group A (14.8 ± 8.1) and Group B (14.4 ± 6.6) were comparable, which was significantly higher than that of Group C (10.4 ± 4.0) (*p* < 0.001) (Figure 4b).

The physical component summary score of SF-8 of FD (43.1 ± 8.5) was comparable to that of Group A (42.5 ± 7.3) and Group C (45.9 ± 6.3), which was significantly higher that of Group B (46.4 ± 6.8). Furthermore, the mental component summary scores among four groups were not significantly different (Figure 4c,d).

## 4. Discussion

This is a multicenter, prospective, observational study to investigate the clinical significance of dyspeptic patients who do not meet Rome IV criteria as well as usefulness of the criteria. The present study demonstrated that approximately only 60% of the patients visiting hospitals with complaints of dyspeptic symptoms met Rome IV diagnostic criteria for FD. The 126 patients in FD group were subdivided into three groups. There were 22 (17%), 79 (63%), and 25 (20%) patients in EPS, PDS, and overlap, respectively. A previous study from Japan by Manabe et al. [25] reported the ratio of 54.9%, which is comparable with our report. On the other hand, a recent Belgian study reported that approximately 90% of such patients met the Rome criteria, which was far greater than that described in our study [7], while an Indian study and a Chinese study reported 21.7% and 20.2% of such patients met the Rome criteria, respectively [13,14], which was further smaller than that of ours and the Belgium study. Although the reason for the difference of the ratio according to the different countries is not known, environmental factors including dietary habits [26,27], sociocultural factors, or genetic factors [28] may be involved. 

There are many dyspeptic patients who do not meet Rome criteria, at least in Asian countries, health related quality of life (HR-QOL) of such patients has not been well documented. In this study, we compared QOL among three groups (FD, non-FD and healthy subjects) and found that the QOL of FD and non-FD patients was significantly decreased compared to healthy subjects, suggesting QOL was significantly impaired even in non-FD patients. This clearly demonstrates the need for therapeutic intervention in patients complaining of dyspeptic symptoms regardless of whether they meet the diagnostic criteria. On the other hand, HR-QOL as well as GSRS and HADS was less impaired in non-FD patients compared to FD patients.

Accordingly, we investigated what part of the Rome criteria brings the difference of QOL between FD and non-FD patients. In this study, the non-FD patients were further subdivided into three groups (Group A, B, C) to compare the difference of HR-QOL. The Group A patients met the frequency criteria of symptom (several times a week), but not met the duration criteria (longer than six months). The Group B patients did not meet the frequency criteria and met the duration criteria. Our results showed GSRS, HADS and HR-QOL of FD were comparable with that that of group A, while they were significantly different compared to group B and C. As group A was defined to meet the frequency criteria (not meet duration criteria), this finding suggests symptom frequency, not duration, has significant impact on the QOL of the patients. In other words, it is clinically important to reduce the frequency of dyspeptic symptoms to improve patients’ QOL.

On the contrary, Kinoshita et al. reported non-FD patients cannot be labeled as FD due to duration criteria in Japan [29]. They suggested this is largely depended on the easy access to the hospital in our country, which might be associated with Japanese universal insurance system. Unlike Kinoshita’s study, in our study, only 20% of non-FD patients did not meet the duration criteria and 60% did not meet the frequency criteria. We speculate the reason for this discrepancy is their study population was patients in primary care clinic, and that in our study was those mainly in tertiary care university hospital, meaning many of them being referred patients. As gastric cancer is the leading malignancy in Japan, the people tend to visit primary care clinic once epigastric pain and discomfort develop, owing to the fear of gastric cancer [30]. This may explain the difference between the ratio of Kinoshita’s study and ours. In fact, other study from Japan reported the ratio dyspeptic subjects meeting Rome criteria was 10.3% [15].

There are several limitations in this study. First, 71% of patients participating this study was from a tertiary care university hospital, suggesting many of them were referred patients, which may bring the selection bias. Another limitation is that dividing non-FD patients into three subgroups (Group A, B and C) resulted in the relatively small number of patients in each subgroup, which may have brought type II error. 

## 5. Conclusions

The findings of this study indicate that approximately 40% of those visiting hospitals with dyspeptic symptoms do not meet the Rome IV criteria, and HR-QOL of such patients is clearly impaired. Given that an increased frequency of symptoms may impact on impairment of a patient’s QOL, the duration criteria of the Rome IV definition is likely to be reconsidered.

## Figures and Tables

**Figure 1 jcm-11-00021-f001:**
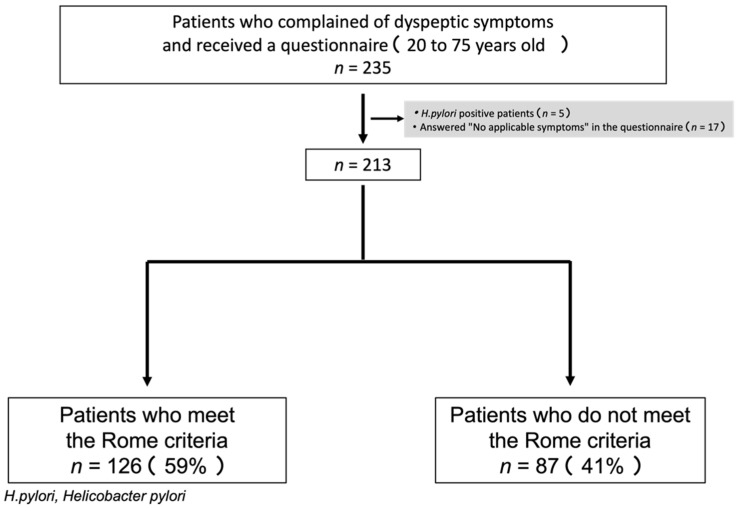
Consort chart.

**Figure 2 jcm-11-00021-f002:**
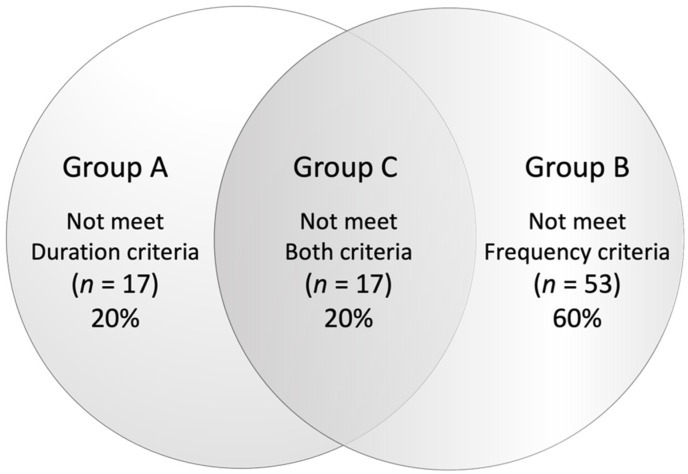
Subdivision of non-FD patients.

**Figure 3 jcm-11-00021-f003:**
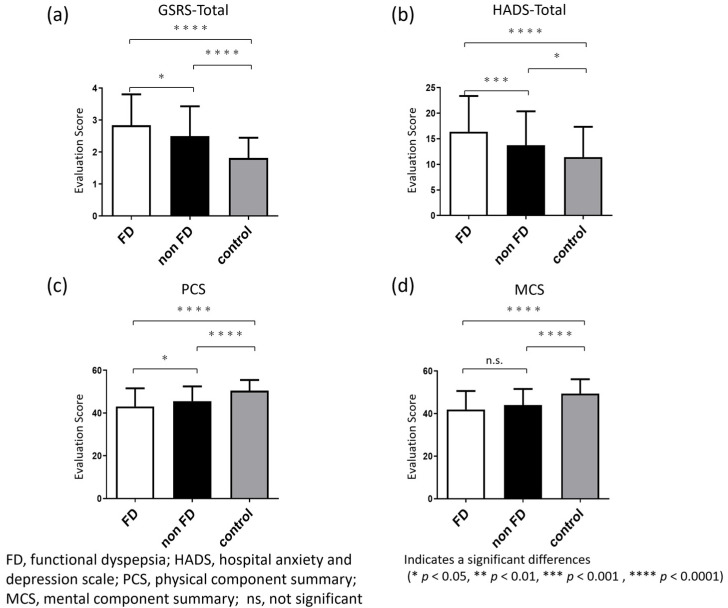
GSRS, psychological and general QOL score of FD and non-FD patients. (**a**,**b**) The total GSRS and HADS score were significantly higher in FD group than non-FD group, which was higher than the control group; (**c**,**d**) Physical and mental component summary scores were significantly lower in FD and non-FD patients than in the control group, and PCS was also significantly lower in the FD group than in the non-FD group. No significant difference was observed between the FD and non-FD groups.

**Figure 4 jcm-11-00021-f004:**
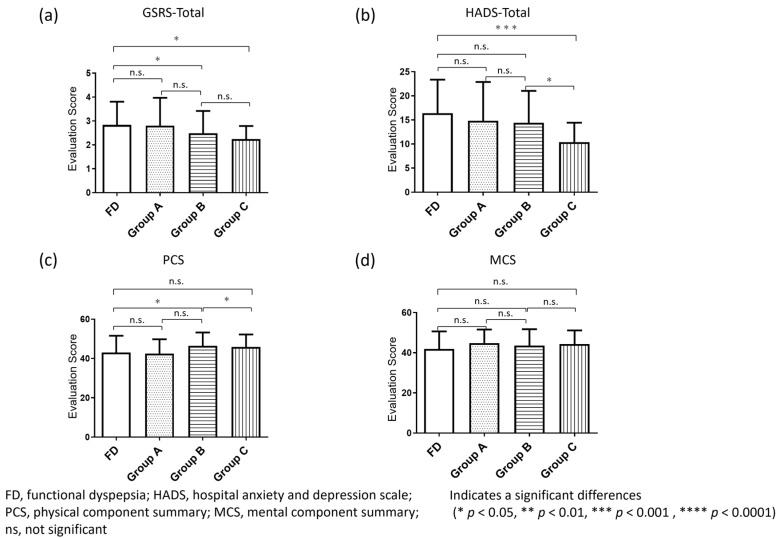
GSRS, psychological and general QOL score in subdivision of non-FD patients. (**a**) The overall scores of the GSRS for Group A and the FD group are comparable, whereas those of Group B and Group C were significantly lower than that of the FD group; (**b**) The overall psychological scores for Group A and Group B are not significantly different from that of the FD group, but the Group C score is significantly lower than that of the FD group; (**c**,**d**) The physical component summary scores of general QOL for Group A and Group C are not significantly different from that of the FD group, but the score of Group B is significantly higher than that of the FD group. Furthermore, the mental component summary scores of Group A, Group B and Group C are not significantly different from that of the FD group.

**Table 1 jcm-11-00021-t001:** Characteristics and symptoms of the patients.

Characteristics and Symptoms	Dyspepsia				Control	*p* Value
	Meet the Rome Criteria (FD Group)			Do Not Meet the Rome Criteria (Non-FD Group)		
	EPS	PDS	Overlap			
Patients (*n*)	22	79	25	87	111	
Age (years), mean ± SD	54.0 ± 16.9	54.1 ± 14.6	55.3 ± 13.7	54.9 ± 12.1	54.4 ± 11.6	0.923
Gender (*n* (% female))	14 (63.6)	56 (70.9)	16 (64.0)	60 (68.8)	74 (66.7)	0.939
BMI (kg/m^2^), mean ± SD	21.5 ± 3.1	21.6 ± 4.2	21.1 ± 3.5	21.8 ± 3.2	22.8 ± 4.2	0.105
Smoking (*n* (%))	4 (18.2)	20 (25.3)	5 (20.0)	17 (19.5)	32 (28.8)	0.567
Drinking (*n* (%))	7 (31.8)	23 (29.1)	6 (24.0)	24 (28.6)	43 (38.7)	0.417
After eradication of *Helicobacter pylori* (*n* (%))	5 (22.7)	16 (20.3)	7 (28.0)	22 (26.0)	29 (26.1)	0.867
Postprandial fullness (*n* (%))	0	48 (60.8)	20 (80.0)	32 (36.8)	0	
Early satiation (*n* (%))	0	41 (51.9)	19 (76.0)	24 (27.6)	0	
Epigastric pain or burning (*n* (%))	22 (100)	30 (38.0)	25 (100)	31 (35.6)	0	
Postprandial epigastric pain or burning (*n* (%))	0	34 (43.0)	0	17 (19.5)	0	

BMI: body mass index, EPS: epigastric pain syndrome, PDS: postprandial distress syndrome, FD: functional dyspepsia.

**Table 2 jcm-11-00021-t002:** Comparison of characteristics among five groups (FD group, Group A, Group B, Group C and control group).

Characteristics	Meet the Rome Criteria	Do Not Meet the Rome Criteria			Control	*p* Value
	FD Group	Group A	Group B	Group C		
Patients (*n*)	126	17	53	17	111	
Age (years), mean ± SD	54.3 ± 14.9	59.2 ± 11.6	54.8 ± 11.8	52.8 ± 14.0	54.4 ± 11.6	0.931
Gender (*n* (% female))	86 (68.5)	11 (64.7)	34 (64.2)	13 (76.5)	74 (66.7)	0.909
BMI (kg/m^2^), mean ± SD	21.8 ± 3.5	21.9 ± 3.2	21.7 ± 3.5	22.2 ± 3.2	22.8 ± 4.2	0.257
Smoking (*n* (%))	29 (23.4)	3 (17.6)	14 (26.4)	2 (11.8)	32 (28.8)	0.516
Drinking (*n* (%))	36 (28.8)	7 (41.1)	17 (32.1)	3 (17.6)	43 (38.7)	0.277

BMI: body mass index, FD: functional dyspepsia.

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
