# Peer review of "Evaluation of QOL in Patients with Dyspeptic Symptoms Who Meet or Do Not Meet Rome IV Criteria"

_jcm, 2021, doi:10.3390/jcm11010021_

Round 1
Reviewer 1 Report
The subject is interesting, and the information generated characterizes a group of patients frequently seen in clinical practice and not usually considered in therapy studies. However, the study has some errors and requires revisions and clarifications of certain points to meet the requirements of the journal.
Major issues
- It would be very useful to include in the analysis and comparisons the distribution and characteristics of the two subsyndromes of patients with a Rome IV diagnosis of functional dyspepsia: Postprandial Distress Syndrome and Epigastric Pain Syndrome; and not only the overall functional dyspepsia group. Comparing these groups with the rest of the non-DF and healthy volunteers would enrich the discussion and conclusions.
Minor issues
-Consider changing the title, as it assumes that everyone had functional dyspepsia, whether or not they met the Roma IV criteria, which is not in line with the detail of the text, in fact it calls a group non-FD patients.
-After a period or when starting a sentence use words and not numbers.
-Detail whether all endoscopies included search for H. pylori by urease test or gastric biopsies.
-Detail objective criteria for item 2 of the exclusion criteria.
-Detail item 3 of the exclusion criteria. Based on symptoms or Lyon consensus endoscopic criteria.
-Detail what criteria were used for exclusion point 4. Rome IV?
-Explain which endoscopic surgeries were included.
-Explain the "other reasons" in point 8 of the exclusion criteria.
-Please detail if the HADS scale is validated in the Japanese population.
-Figure 1: Correct the word "Helicobactor".
-The numbers in the figure do not match the data in the text, specifically group B.
- Improve resolution figures 3, 4 and A2 almost not readable.
Reviewer 2 Report
This study investigated health related quality of life (HR-QOL), gastrointestinal symptoms, and psychological factors in functional dyspepsia (FD) and non-FD patients. It also examined the clinical significance of non-FD patients and usefulness of Rome IV criteria. This study has strengths in terms of the large sample size and originality of comparing FD patients with non-FD and healthy individuals. Some of the things that need to be revised are as follows.
- Please designate primary outcome and explain how the study size was arrived at.
- Correct the numbers in Figure 2 that do not match those in the text.
- Interpretation of the result that symptom frequency, not duration, has significant impact on the QOL of the FD patients should be cautious. It is recommended to add subgroup comparison of clinical characteristics among 5 groups (FD group, Group A, Group, Group C and control group) to secure more evidence.
